# Establishment of a Dataset for the Traditional Korean Medicine Examination in Healthy Adults

**DOI:** 10.3390/healthcare12090918

**Published:** 2024-04-29

**Authors:** Soyoung Kim, Ancho Lim, Young-Eun Kim, Youngseop Lee, Hyeong Joon Jun, Mi Hong Yim, Daehyeok Kim, Purumea Jun, Jeong Hwan Park, Sanghun Lee

**Affiliations:** 1KM Data Division, Korea Institute of Oriental Medicine, Daejeon 34054, Republic of Korea; ssuying1221@kiom.re.kr (S.K.); limancho@kiom.re.kr (A.L.); jade2010@kiom.re.kr (Y.-E.K.); rheey119@kiom.re.kr (Y.L.); jeonhj@kiom.re.kr (H.J.J.); purumea@kiom.re.kr (P.J.); 2Korean Convergence Medical Science, University of Science and Technology, Daejeon 34113, Republic of Korea; 3Digital Health Research Division, Korea Institute of Oriental Medicine, Daejeon 34054, Republic of Korea; mh.yim@kiom.re.kr (M.H.Y.); kdh2440@kiom.re.kr (D.K.)

**Keywords:** digital healthcare, dataset, traditional Korean medicine, traditional Korean medicine examination, protocol

## Abstract

We established a protocol for the traditional Korean medicine examination (KME) and methodically gathered data following this protocol. Potential indicators for KME were extracted through a literature review; the first KME protocol was developed based on three rounds of expert opinions. The first KME protocol’s feasibility was confirmed, and data were collected over four years from traditional Korean medicine (KM) hospitals, focusing on healthy adults, using the final KME protocol. A literature review identified 175 potential core indicators, condensed into 73 indicators after three rounds of expert consultation. The first KME protocol, which was categorized under questionnaires and medical examinations, was developed after the third round of expert opinions. A pilot study using the first KME protocol was conducted to ensure its validity, leading to modifications resulting in the development of the final KME protocol. Over four years, data were collected from six KM hospitals, focusing on healthy adults; we obtained a dataset comprising 11,036 healthy adults. This is the first protocol incorporating core indicators of KME in a quantitative form and systematically collecting data. Our protocol holds potential merit in evaluating predisposition to diseases or predicting diseases.

## 1. Introduction

Qualitatively recorded data based on human sensory perception can be transformed into quantitative metrics through digital technology [1]. Digital healthcare technology, exemplified by smart devices, employs high-performance sensors and computational processing units [2]. These technologies are proxies for the human senses and memory, effectively substituting subjective observations with objective and quantifiable data [3].

Within the framework of a single official national health system (NHS), Korea administers a bifurcated system that separately offers conventional medicine and traditional Korean medicine (KM) services. Specifically, Korean medicine doctors (KMDs), under governmental regulations, deliver healthcare services such as acupuncture, moxibustion, cupping, manipulation, and herbal medicine. However, these services constitute a minor fraction within the Korean NHS [4]. 

Modern medicine, based on quantifiable data, is a prominent advancement in medical technology, mitigating human diseases and extending healthy lifespans [5,6]. In the field of KM, there has been a continuous evolution in technology through the application of cutting-edge techniques. Moreover, the use of large data has become increasingly prominent in healthcare and KM. In particular, the Sasang constitution is a crucial resource in KM that emphasizes the preventive aspects of health related to lifestyle, nutrition, and behavior [7]. Analyzing Sasang constitutional data enables us to understand a patient’s constitutional characteristics and develop personalized KM treatment methods [8,9]. KM emphasizes individualized treatment based on constitutional characteristics, and leveraging such data can greatly assist in finding optimal treatment methods tailored to each patient’s unique constitution [10]. Recently, the Sasang constitution multicenter bank for Koreans has been continuously built quantitatively [11]. However, limitations still exist in the field of KM owing to the scarcity of structured and quantitative data. 

In this study, we developed a traditional Korean medicine examination (KME) protocol by standardizing and quantifying the measurement protocol based on the core indicators of KM commonly used to evaluate the health status of patients in KM clinical practice, and systematically collected data. The KME protocol holds potential merit in assessing the risk of diseases, or in evaluating the anticipation of diseases. In addition, the KME dataset aimed to explore the potential of using large data for KM. 

## 2. Materials and Methods

The workflow comprised the following specific steps (Figure 1).

### 2.1. Preliminary Review

A comprehensive list of potential core indicators for KME was compiled by three KMDs through a literature review encompassing several sources. This included an analysis of published KM textbooks, the current status of KM questionnaires or device development, KM diagnostic questionnaires and developed devices, such as those for tongue and pulse diagnosis, as quantitative measurements via the Oriental Medicine Advanced Searching Integrated System (OASIS) database (https://oasis.kiom.re.kr/index.jsp (accessed on 18 May 2018)), clinical research protocols for projects conducted by the Korea Institute of Oriental Medicine (KIOM), and the status of domestically available commercial medical devices.

### 2.2. Expert Opinions

The panel participating in the two rounds of the survey comprised 15 KMDs recommended by the Society of Korean Medicine and more than ten diverse specialty societies within the Association of Korean Medicine. The first survey assessed whether the nine primary categories and 39 subcategories identified in the preliminary review, along with the 175 indicators contained within them, were suitable candidates for KME core indicators. 

To assess the suitability of the categorized core indicators for KME, a secondary survey was conducted among the same experts as in the first round. Each indicator was evaluated on a 3-point Likert scale: 1 = unsuitable, 2 = suitable, and 3 = very suitable. The criterion for indicator selection was receiving an average expert response score of 2.5 or higher on a three-point scale. These core indicators were bifurcated into questionnaire categories, such as thirst and xerostomia, and measurement categories such as halitosis. To assist experts in their evaluation, validated questionnaires, standardized assessment protocols, proxy indicators, and exemplar units of measurement were proposed for each category. Additionally, indicators with an average score of less than 2.5 were reconsidered through internal discussions among the researchers under three conditions: appropriate measures were already available in the clinical practice of KM, such as body shape/posture, halitosis, and electroacupuncture according to Voll (EAV); the proportion of respondents with a score of 2 or higher was greater than 80%; or they were deemed meaningful when collected together, such as type (2.4), frequency (2.5), and duration (2.5) of exercise. Finally, by predetermined criteria, the chosen core indicators were simply incorporated into the KME indicators. 

In the third round, the final selection of the questionnaire and measurement methods for the first KME protocol development was based on the indicators identified through the secondary expert survey. The indicators were divided into questionnaire and medical examination categories. The task involved verifying whether the indicators could be assessed in a real-world clinical setting. Additionally, it encompassed the process of choosing a method or device capable of quantitatively evaluating these indicators and conducting preliminary measurements. At this stage, four biomedical engineers and six KMDs were consulted. The six KMDs included three specialists in KM diagnosis, one specialist in KM neuropsychiatry, one specialist in KM internal medicine, and one specialist in KM gynecology. For areas requiring expert opinion, such as color quantification through photographs or quantification of dietary intake and eating habits, additional consultations were conducted with a professional photographer and a professor of nutrition science. 

### 2.3. Pilot Study Using the First KME Protocol

The development of the first KME protocol was based on the core indicators of KME. To evaluate the feasibility of this study, a pilot study was conducted by 7 KMD experts using the developed protocol. Through the pilot study, several points were improved, and the final protocol was confirmed.

### 2.4. Final KME Protocol Development and Update

The final KME protocol was implemented in 2020 and updated until 2023. These updates aim to prevent response errors and enhance measurement accuracy during the data collection and cleaning processes.

### 2.5. Establishment of Electronic Case Report Form System

The mobile application software used for completing the questionnaires and the electronic case report form (eCRF) were developed in compliance with the relevant guidelines provided by the Ministry of Food and Drug Safety [12]. The questionnaire completion software is compatible with both Android and iPhone operating systems, whereas the eCRF has been optimized for the Chrome web browser, enabling efficient recording and management of all data collected during clinical trials. Clinical research participants completed all questionnaire information directly through a mobile application. All participants were required to thoroughly complete the entire questionnaire through a dedicated mobile application before proceeding to the medical examination phase of the KME program.

The app automatically processed the data securely before transferring them to the server. At each hospital, designated researchers entered medical examination data using the eCRF system. In turn, clinical research associates ensured data accuracy through ongoing monitoring. The data obtained from eCRF were processed using R version 4.2.1 (R Foundation for Statistical Computing, Vienna, Austria).

### 2.6. Dataset Collection

This study collected detailed observational data. The KME population comprised non-institutionalized, healthy adults residing in Korea. This study was conducted at the following six sites: Naju Dongshin University Korean Medicine Hospital, Pusan National University Korean Medicine Hospital, Gachon University Gil Medical Center, Dongguk University Ilsan Oriental Hospital, Dunsan Korean Medicine Hospital of Daejeon University, and Semyung University Korean Medicine Hospital. The study was conducted in each hospital with the same KME protocol and standard operating procedures (SOPs) every year. All hospitals were furnished with the same medical measuring devices, which were tested and validated. To prepare for the clinical study, we conducted three rounds of training following a standardized SOP. Furthermore, to ensure protocol compliance, we performed regular monitoring visits. The device data uploaded to the eCRF were monitored in-house in real time. We conducted an audit to guarantee the reliability of the data. All participants, regardless of sex, were voluntarily recruited on a first-come first-served basis from those who completed the national general health screening. All participants were adults aged ≥19 years old with no cognitive impairment. The exclusion criteria were as follows: inability to move independently or use measuring devices, such as an InBody device, and current diagnosis of the following: cardiovascular diseases (e.g., myocardial infarction, congestive heart failure, angina, and arrhythmia), cerebrovascular diseases (e.g., cerebral infarction and paralysis), malignant neoplasms (i.e., cancer), mental illnesses (e.g., depression and anxiety disorder), rheumatoid arthritis, and thyroid diseases (e.g., hyperthyroidism and hypothyroidism). This study was conducted in accordance with the Declaration of Helsinki and was approved by the Institutional Review Board of each of the six hospitals assessed. Informed consent was obtained from all the participants involved in the study.

## 3. Results

### 3.1. Preliminary Review

Initially, we examined the status of the questionnaire development for KM diseases according to the Korean Standard Classification of Diseases. There were 32 questionnaires for pattern identification of diseases or symptoms, 9 for Sasang constitution diagnosis, 10 for health assessment, and 4 for treatment evaluation. Subsequently, we scrutinized the quantification tools for diagnostic indicators in KM textbooks. In the four core diagnostic examinations, 20 indicators for inspection, 7 indicators for listening and smelling, 6 indicators for inquiry, and 3 indicators for palpation were identified. Moreover, we examined eight research projects conducted by the KIOM, revealing the utilization of 12 types of equipment tests. These include pulse tonometry devices, tongue image analysis systems, abdominal examinations, heart rate variability measurements, and electroencephalography. Finally, among domestic commercial medical devices, 224 types of diagnostic or measuring devices were identified as potential candidates for use in KME. Through a review of research data from the three KMDs, 175 potential core indicators for KME were categorized into 39 subcategories under 9 primary categories: daily living symptoms, health habits, subjective symptoms, health status, KM diagnosis, laboratory results, medical information, DNA analysis, and menstrual health.

### 3.2. The First, Second, and Third Expert Opinions

Through the first round of expert opinions on the suitability of KME core indicator candidates, the nine primary categories were condensed into four: daily living symptoms, health habits, menstrual health, and KME diagnosis. The experts’ main opinion was that categories such as subjective symptoms, health status, and DNA analysis should be considered key indicators for each disease rather than KME core indicators. The 39 subcategories were reduced to 22; newly added subcategories included body shape/posture and facial color under daily living symptoms, iris diagnosis, and EAV under KM diagnosis. Finally, the total number of indicators was reduced from 175 to 98.

The indicators selected from the second round of expert opinions were classified into two categories: questionnaires and measurements. The questionnaire categories included thirst/xerostomia/amount of water consumed, digestion, stool, urine, sleep, sweating, heat and cold patterns, emotions, stress, eating habits, alcohol habits, smoking habits, exercise habits, and menstrual health status. The measurement category encompassed body shape/posture (3D body scan), facial color, halitosis, digestion (electrogastrogram, EGG), urine (chemical ingredient, color), stress (heart rate variability, HRV), heat and cold patterns (resting metabolic rate, RMR; ventilation rate, VR), height/weight, body composition, blood pressure, body heat (thermography), pulse diagnosis, tongue diagnosis, abdominal examination, and Sasang constitution. In this round, the total number of indicators was reduced from 98 to 86.

In the third round, we aimed to restructure the method of measuring indicators in an actual clinical setting using questionnaires and medical examinations. Some indicators that were difficult to implement in the field of KM were excluded. For example, although the state of gastrointestinal movement is an indicator, the electrogastrography suggested by experts to measure it could not be legally used in the field of KM. Through this process, the total number of indicators was reduced from 86 to 73. Furthermore, some indicators were included in both the questionnaire and medical examination. For instance, stress was selected to be measured by both the Perceived Stress Scale in the questionnaire and by HRV in the medical examination. Indicators related to heat and cold patterns were duplicated in the questionnaire and measured with RMR and VR through Quark RMR indirect calorimeter. Finally, all the information measured by the device we selected in the medical examination was included in the measurement items. For example, during the chemical analysis of urine, not only the items related to urine color such as urobilinogen and bilirubin, but also additional incoming items such as glucose, urine specific gravity, ketone body, occult blood, pH, and urine nitrite were included.

The KME program was divided into two categories: questionnaires and medical examinations. The questionnaire categories included thirst/xerostomia/amount of water consumed, digestion, stool, urine, sleep, sweating, heat and cold patterns, eating habits, menstrual health, alcohol habits, smoking habits, exercise habits, Sasang constitution questionnaire, Pittsburgh Sleep Quality Index (PSQI), Perceived Stress Scale (PSS), and Core Seven Emotions Inventory (CSEI). The medical examination category comprised on-site measurement equipment tests, including chemical analysis of urine, urine color, body temperature, skin moisture, and lipids, tongue diagnosis, salivary flow rate, halitosis, pulse diagnosis, height/weight, body composition, HRV, RMR, VR, abdominal examination, blood pressure, Sasang constitution measurement, 3D body scan, digital thermography, and multiple allergen simultaneous test (MAST) (Table 1). The questionnaire category was systematically refined for readability and participant convenience, and SOPs were developed for each measurement method.

### 3.3. Pilot Study Using the First KME Protocol

We developed an inaugural protocol for KME, encompassing parameters such as KME spatial prerequisites, device installation milieu, SOPs for questionnaire and device measurement, stipulations for participant adherence, and requirements for examiners. A pilot study was conducted with 90 participants at 3 KM hospitals in 2019 [13]. For the study, health check-up data were added to the final protocol to complement the study’s results and confirm the participant’s health status more objectively. To increase the validity of the survey content, the indicator ‘exercise habits’ was changed to the International Physical Activity Questionnaire Short Form (IPAQ-SF), and the CSEI 100-item questionnaire was changed to the 28-item Core Seven Emotions Inventory-Short Form (CSEI-S) to increase the subject’s understanding and ease of response. Finally, active oxygen and posture analyses were added to the medical examinations. All experts who participated in the pilot study concurred with the revised KME protocol.

### 3.4. Final KME Protocol

The final KME protocol is presented in Table 2 [14]. The protocol consisted of the demographics of the participants, 16 questionnaires, 20 medical examinations, and the MAST. 

Participant demographic information included characteristics, medical history, allergy history, and health checkup data. Participants were asked to complete 16 questionnaires through the application, including 11 clinical questionnaires and 5 previously developed questionnaires. The clinical questionnaire consisted of 11 items: thirst/xerostomia/amount of water consumed; digestion, stool, urine, sleep, sweating, heat, and cold patterns; eating habits; menstrual health; alcohol consumption; and smoking habits. In addition, the PSQI [15,16], PSS [17,18], CSEI-S [19], IPAQ-SF [20,21], and Sasang constitution Questionnaire [22] were used to measure sleep quality, perceived stress, emotion, exercise, and Sasang constitution. The 20 medical examinations were as follows: (1) chemical analysis of urine, (2) active oxygen, (3) urine color, (4) body temperature, (5) skin moisture and lipids, (6) tongue diagnosis, (7) salivary flow rate, (8) halitosis, (9) pulse diagnosis, (10) height/weight, (11) body composition, (12) heart rate variability, (13) resting metabolic rate, (14) ventilation rate, (15) abdominal examination, (16) blood pressure, (17) Sasang constitution measurement [20], (18) 3D body scan, (19) posture analysis, and (20) digital thermography. In addition, participants who reported allergies underwent the MAST [23].

### 3.5. The KME Protocol Updates

Table 3 shows the content updated annually while studying the KME protocol. Lifestyle-related diseases and self-rated health were added to the participants’ demographic information for data analysis based on the health status level of the participants, starting in 2022. Lifestyle-related diseases included the diagnosis of hypertension, diabetes, and hyperlipidemia, along with their diagnosis time and current prevalence [24]. Self-rated health was defined as the degree of health the participants considered themselves to be in on a 5-point Likert scale [25]. During the medical examination, the posture analysis, 3D body scans, and digital thermography were altered. For posture analysis, the direction of sight was added in 2021 and 2022; however, from 2023 onward, all posture analysis measurements were removed from the protocol. Concurrently, the 3D body scan was also removed in 2023. This decision was based on a preliminary analysis conducted with data collected up until 2022, which indicated that both the 3D body scan and posture analysis had low data utilization. Digital thermography added peripheral body temperature measurements of the palm, back of the hand, and sole starting in 2023. MAST has been updated to test 108 types since 2023, with 15 types which people have recently been encountered frequently with added to the existing 93 types.

### 3.6. Status of Dataset Collection

An analysis of the demographic characteristics, physical features, and habits of 11,036 healthy adults was conducted over three years, segmented by year. The age distribution revealed that the majority of participants were in their 40s. In terms of sex distribution, 70.9% were female and 29.1% were male. The participants exhibited an average height of 163 ± 8.28 cm and an average weight of 63.5 ± 12.3 kg. Among the participants, 80.6% were non-smokers, and 64.7% were alcohol consumers (Table 4).

## 4. Discussion

To standardize the field of Korean medicine (KM), this study aimed to develop a protocol for KME and systematically collect data according to established KME procedures. Our study focused on a population of healthy adults, resulting in a comprehensive dataset comprising 11,036 individuals. This was an observational study conducted on a voluntary basis, accepting participants on a first-come, first-served basis without any sex distinction. However, the results indicated a higher participation rate among females. The initial target size for the participants was 12,000, but this was downsized due to COVID-19, ultimately comprising 11,036 individuals. 

The process of standardization, aimed at achieving an optimal degree of order, led to the creation of a universally comprehensible common system [26]. High-quality quantitative clinical data characterized by adherence to stringent measurement standards were collected [27]. The process involved specifying the physical quantity under measurement using standardized tools and methods [28] ensuring consistency, and mitigating the risk of error. Collated data were then preserved in a standardized format, maintaining their integrity and original form. Establishing an ecosystem for the systematic collection of real-world data from KM clinical procedures is critical. This methodological approach not only ensures data reliability but also propels the scientific advancement of KM.

In the process of establishing the KME protocol, we encountered several challenges and have taken measures to address them. Firstly, the need to minimize device malfunction was identified. Given the large number of participants measured, it is crucial to ensure that there are no measurement errors caused by the devices utilized. To prevent such errors, we used sufficiently validated equipment and deployed dedicated personnel at each site to immediately resolve any device errors. Additionally, regular staff training according to the SOPs enhanced the work capabilities of clinical coordinators [13]. Secondly, participant satisfaction in KME was of utmost importance. To augment this satisfaction, it is vital to provide participants with their results. Additionally, the analysis of examination results and the interpretation of KM diagnosis to participants were essential components of this process. These experiences provide valuable insights into the practical considerations and potential pitfalls in the implementation of a KME protocol and may serve as a guide for future research in this area.

The Tongue Diagnosis Data Center was established to conduct standardized research on diseases using data collected through tongue diagnosis, which is a component of KME programs. The center was designated the 63rd National Reference Standards Data Center in Korea by the Korean Agency for Technology and Standards, effective from 19 January 2023. In general, reference standards refer to measures officially validated through scientific analysis and evaluation of the accuracy and reliability of the measured data and information.

The traditional subjective expression of a patient’s pulse being weak has been transformed into an objective context with the establishment of these data. This step enables the provision of an objective indicator stating that, for example, ‘Your pulse corresponds to the lower 20% of the pulse strength standard of Korean women in their 40s’. This approach quantifies the traditional qualitative aspects of pulse diagnosis and enhances precision and reproducibility.

The significance of this study lies in its pioneering approach to data collection in KM. It establishes a definitive set of information deemed essential for data collection during clinical practice, standardizes the data collection process, and enhances the reliability and applicability of the data for further research and clinical studies. Therefore, it is essential to create a comprehensive dataset for healthy individuals. This dataset served as a standardized reference, enabling a comparative analysis of the current condition of the patients. The ultimate aim is to use this extensive dataset to derive insights into patient conditions, thereby transforming the traditionally subjective patient status into an objective, quantifiable metric. Given the relative simplicity of data collection from a healthy population, our primary emphasis was on the rapid construction of a robust reference database for this demographic. At the same time, if we methodically gather disease-related data for our healthy participants, a comprehensive and scientifically rigorous approach to data collection and analysis will be ensured. This strategy not only bolsters the precision and reliability of our data but also significantly contributes to the progression of medical research and patient care.

However, this study has some limitations. The protocol, initially intended for a healthy population, has not been validated for use with specific disease populations, which limits its reliability as a standard. Amassing quantitative data from patients diagnosed with specific diseases is critical. However, without a comparative analysis involving a dataset from diseased individuals, the protocol’s suitability for these populations is uncertain. Furthermore, despite the collection of unstructured data such as images and sounds, in addition to structured data, a network environment based on hyperconnectivity drives the need to rapidly collect and integrate large volumes of both structured and unstructured data. This holistic approach to data collection ensures the creation of comprehensive datasets that encapsulate diverse forms of information, thereby enhancing the depth and breadth of the analysis [29]. Moreover, the items evaluated through questionnaires are not continuous variables, which may impose limitations on the analytical methods and outcomes. Additionally, the potential impact of lifestyle-related diseases such as hypertension, diabetes, and hyperlipidemia on the test indicators must be considered during data analysis, which we only incorporated starting half-way through the project.

## 5. Conclusions

This study represents a significant step toward the standardization of KM, the development of a protocol for KME, and the systematic collection of data by KMEs’ standard operating procedures. In contemporary medical practice, the cornerstone of diagnosis lies in the comparison of a patient’s diagnostic indicators with the average range established for a healthy population. However, in the realm of KM, the absence of physiological definitions for certain concepts has hindered the development of standard datasets for patient anomalies and average values for healthy individuals. This study is pioneering in its approach to standardize the clinical diagnostic practices of KMDs, moving away from reliance on subjective senses.

Currently, the absence of a standardized diagnostic protocol means that medical data in KM are not being stored as digitized data. As the standardized KME protocol and the dataset become more prevalent, and as sufficient data on the normal values for healthy individuals and anomalies in patients accumulate, we will be poised to embrace the era of digital medicine. The prognostic information derived from the diagnosis and treatment in Korean medicine could be standardized and transformed into big data. This development will foster a favorable environment for the integration of artificial intelligence into KM. Despite its analog nature, which has previously been deemed a barrier, this approach paves the way for potential breakthroughs in the field of KM. Overall, this data collection approach in the KM field sets a new standard, paving the way for future research and clinical studies.

## Figures and Tables

**Figure 1 healthcare-12-00918-f001:**
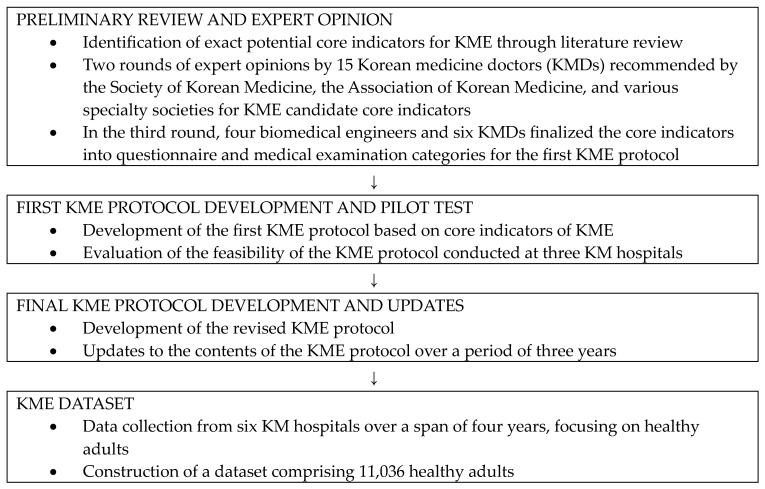
Overview of the flow chart.

**Table 1 healthcare-12-00918-t001:** Three rounds of expert opinions for traditional Korean medicine examination protocol.

Preliminary Review	Nine Categories (Daily Living Symptom, Health Habits, Subjective Symptoms, Health Status, KM Diagnosis, General Health Screen, Medical Information, DNA Analysis, Menstrual Health), 39 Subcategories, 175 Indicators				
1st expert opinion	Category	Daily living symptom	Health habits	Menstrual health	KM diagnosis
	Subcategory	Body shape/posture, facial color thirst/xerostomia/amount of water consumed, halitosis, digestion, stool, urine, sleep, sweating, heat and cold patterns, emotion	Eating habits, alcohol habits, smoking habits, exercise habits	Menstrual health	Pulse diagnosis, tongue diagnosis, abdominal examination, Sasang constitution, iris diagnosis, EAV
2nd expert opinion	Category	Questionnaire		Measurement	
	Subcategory	Thirst/xerostomia/amount of water consumed, digestion, stool, urine, sleep, sweating, heat and cold patterns, emotion, stress, eating habits, alcohol habits, smoking habits, exercise habits, menstrual health		Body shape/posture (3D body scan), facial color, halitosis, digestion (EGG), urine (chemical ingredient, color), stress (HRV), heat and cold patterns (RMR, VR), height/weight, body composition, blood pressure, body heat (thermography), pulse diagnosis, tongue diagnosis, abdominal examination, Sasang constitution	
3rd expert opinion	Category	Questionnaire		Medical examination	
	Subcategory	Thirst/xerostomia/amount of water consumed, digestion, stool, urine, sleep, sweating, heat and cold patterns, eating habits, menstrual health, alcohol habits, smoking habits, exercise habits, Sasang constitution questionnaire, PSQI, PSS, CSEI		Chemical analysis of urine, urine color, body temperature, skin moisture and lipids, tongue diagnosis, salivary flow rate, halitosis, pulse diagnosis, height/weight, body composition, HRV, RMR, VR, abdominal examination, blood pressure, Sasang constitution measurement, 3D body scan, digital thermography, MAST	

Acronyms: EAV, electroacupuncture according to Vol; EGG, electrogastrography; HRV, heart rate variability; RMR, resting metabolic rate; VR, ventilation rate; PSQI, Pittsburgh sleep quality index; PSS, perceived stress scale; CSEI, the core seven-emotions inventory; MAST, multiple allergen simultaneous test.

**Table 2 healthcare-12-00918-t002:** The final protocol including the core indicators of traditional Korean medicine examination.

Category	Subcategory	Measuring Items	Related Core Indicator
General characteristic	Sex, dominant hand, occupation, education, marital status, number of cohabitants, menstruation status *, concomitant medicine, medical history, allergy history, health checkup data		
Questionnaire	Thirst/xerostomia/amount of water consumed	Thirsty, xerostomia, amount and frequency of drinking water	Thirsty, xerostomia, amount and frequency of drinking water
	Digestion	Appetite, abdominal bloating, abdominal distention, early satiation, food desire during satiety, heartburn, epigastric stiffness, nausea, fatigue after eating	Appetite, abdominal bloating, abdominal distention, early satiation, food desire during satiety, heartburn, epigastric stiffness, nausea, fatigue after eating
	Stool	Number of stools per day, amount, shape, color, discomfort during defecation, odor	Number of stools per day, amount, shape, color, discomfort during defecation, odor
	Urine	Number of urinations per day, color, characteristic, residual urine, nocturia, pain during urination	Number of urinations per day, color, characteristic, residual urine, nocturia, pain during urination
	Sleep	Time from alarm to actual wake up, condition when waking up, insomnia caused by dreams	Time from alarm to actual wake up, condition when waking up, insomnia caused by dreams
	Sweating	Amount of sweating, feeling after sweating, sweat area	Amount of sweating, feeling after sweating, sweat area
	Heat and cold patterns	Cold or heat intolerance, favorite drinks by season, cold or heat sensation	Cold or heat intolerance, favorite drinks by season, cold or heat sensation
	Eating habits	Number of meals per day, meal time, amount of food by meal	Number of meals per day, meal time, amount of food by meal
	Menstrual health *	Obstetrical history, menstrual pattern, menstrual cramps, menstrual blood color, experience with contraception	Obstetrical history, menstrual pattern, menstrual cramps, menstrual blood color, experience with contraception
	Alcohol habits	Type of alcohol, amount of alcohol, drinking frequency	Drinking status, type of alcohol, amount of alcohol, drinking frequency
	Smoking habits	Smoking status, amount of smoking, current smoking status, smoking cessation period, amount smoked before quitting	Smoking status, amount of smoking, current smoking status, smoking cessation period, amount smoked before quitting
	Sasang constitution questionnaire	Personality, meal, digestion, perspiration, excrement, urine, cold and heat, water consumption, sleep, symptoms	Sasang constitution
	PSQI		Sleep–wake cycle, sleep duration, sleep disturbances, awakening incidents, sleep-related discomfort, sleep quality
	PSS		Stress
	CSEI-S		Emotion
	IPAQ-SF		Frequency, duration and intensity of physical activity
Medical examination	Chemical analysis of urine	Urobilinogen, glucose, bilirubin, ketone body, urine specific gravity, occult blood, pH, urine nitrite, leucocytes	Color of urine
	Active oxygen	Degree of active oxygen	Stress
	Urine color	Color of urine	Color of urine
	Body temperature	Measurement of tympanic temperature	Peripheral body temperature
	Skin moisture and lipids	Moisture and lipids of the forehead, upper arm and forearm (both side)	Amount of sweating, sweating area
	Tongue diagnosis	Pale red tongue, tongue fur, bluish-purple tongue, teeth-marked tongue	Tongue diagnosis
	Salivary flow rate	Salivary flow rate	Xerostomia
	Halitosis	Hydrogen sulfide, methyl mercaptan	Halitosis
	Pulse diagnosis	Number, speed, intensity, shape, and depth of pulse (both sides)	Pulse diagnosis
	Height/weight	Height, weight, BMI	Body shape, obesity
	Body composition	Body water, skeletal muscle mass, body fat mass, body fat percentage	Body shape, obesity
	Heart rate variability	TP, LF, HF, VLF, LF/HF ratio	Stress
	Resting metabolic rate	Resting metabolic rate	Cold or heat intolerance
	Ventilation rate	Ventilation rate	Cold or heat intolerance
	Abdominal examination	Pressure pain threshold and pressure depth at the abdominal CV4, CV12, CV14 acupoints and deltoid muscle (reference)	Abdominal examination
	Blood pressure	Systolic and diastolic pressure, and pulse rate (both sides)	Pulse diagnosis
	Sasang constitution measurement	Facial images, body shape, voice	Sasang constitution
	3D body scan	Volume and circumference of body (forehead, neck, axillary, chest, rib, waist, pelvic, hip)	Body shape
	Posture analysis	Distance from the midline to the front of the body (ear, coracoid process, anterior superior iliac spine) and the side of the body (ear, shoulder, knee)	Body posture
	Digital thermography	Mean, standard deviation, minimum and maximum body temperature of front and back side	Cold or heat intolerance
Blood test	MAST **	93 allergens	

* Only for female participants; ** Only participants who reported allergies; Acronyms: PSQI, Pittsburgh Sleep Quality Index; PSS, Perceived Stress Scale; CSEI-S, core seven-emotion inventory—short form; IPAQ-SF, international physical activity questionnaire short form; pH, potential of hydrogen ions; BMI, body mass index; TP, total power; LF, low frequency; HF, high frequency; VLF, very low frequency; CV, conception vessel; MAST, multiple allergen simultaneous test.

**Table 3 healthcare-12-00918-t003:** Annual updates of the traditional Korean medicine examination protocol.

Category	Subcategory	Item	Update Detail	Year (20-)			
				20	21	22	23
General characteristics	lifestyle-related disease	Hypertension, diabetes, hyperlipidemia	Newly added from 2022			√	√
	Self-rated health	Self-rated health	Newly added from 2022			√	√
Medical examinations	Posture analysis	Direction of sight from the midline to the front of the body (ear, coracoid process, anterior superior iliac spine)	Collected only in 2021 and 2022		√	√	
	3D body scan	Volume and circumference of body (forehead, neck, axillary, chest, rib, waist, pelvic, hip)	Removed in protocol from 2023	√	√	√	
	Digital thermography	Mean, standard deviation, minimum and maximum body temperature of palm, sole, and back of hand	Peripheral body temperature measurement added from 2023				√
Blood test	MAST 93	Basic allergen (31 types), food allergen (31 types) inhalant allergen (31 types)		√	√	√	
	MAST 108	Mucor racemosus, rhizopus nigricans, egg yolk, scallop, white bean, hazelnut, Brazil nut, cashew nut, macadamia nut, coconut, carrot, elm, false ragweed, lamb’s quarter, cocklebur	The number of test antigens extended from the existing 93 types of allergens to 108 types starting from 2023				√

**Table 4 healthcare-12-00918-t004:** Characteristics of the traditional Korean medicine examination participants.

Characteristics			2020 (N = 2178)	2021 (N = 2885)	2022 (N = 2986)	2023 (N = 2987)	Total (N = 11,036)
Demographic	Sex						
		Female	1492 (68.5%)	1974 (68.4%)	2146 (71.9%)	2213 (74.1%)	7825 (70.9%)
		Male	686 (31.5%)	911 (31.6%)	840 (28.1%)	774 (25.9%)	3211 (29.1%)
	Age						
		20–29	575 (26.4%)	523 (18.1%)	431 (14.4%)	412 (13.8%)	1941 (17.6%)
		30–39	343 (15.7%)	455 (15.8%)	502 (16.8%)	443 (14.8%)	1743 (15.8%)
		40–49	538 (24.7%)	678 (23.5%)	804 (26.9%)	770 (25.8%)	2790 (25.3%)
		50–59	427 (19.6%)	633 (21.9%)	765 (25.6%)	760 (25.4%)	2585 (23.4%)
		60+	295 (13.5%)	596 (20.7%)	484 (16.2%)	602 (20.2%)	1977 (17.9%)
	Education level						
		High school graduate or below	772 (35.4%)	1118 (38.8%)	1049 (35.1%)	1119 (37.5%)	4058 (36.8%)
		College graduate	261 (12.0%)	323 (11.2%)	415 (13.9%)	331 (11.1%)	1330 (12.1%)
		University graduate	923 (42.4%)	1193 (41.4%)	1317 (44.1%)	1339 (44.8%)	4772 (43.2%)
		In graduate school or above	222 (10.2%)	251 (8.7%)	205 (6.9%)	198 (6.6%)	876 (7.9%)
	Occupation						
		Managers/professionals and related workers	491 (22.5%)	484 (16.8%)	609 (20.4%)	514 (17.2%)	2098 (19.0%)
		Clerks	357 (16.4%)	483 (16.7%)	510 (17.1%)	486 (16.3%)	1836 (16.6%)
		Service workers/Sales workers	228 (10.5%)	316 (11.0%)	315 (10.5%)	361 (12.1%)	1220 (11.1%)
		Skilled workers, e.g., agricultural, manual labor, etc.	578 (26.5%)	765 (26.5%)	608 (20.4%)	643 (21.5%)	2594 (23.5%)
		Unemployed	524 (24.1%)	837 (29.0%)	944 (31.6%)	983 (32.9%)	3288 (29.8%)
	Marital status						
		Never been married	786 (36.1%)	813 (28.2%)	763 (25.6%)	688 (23.0%)	3050 (27.6%)
		Married	1320 (60.6%)	1989 (68.9%)	2109 (70.6%)	2203 (73.8%)	7621 (69.1%)
		Widowed/divorced	72 (3.3%)	83 (2.9%)	114 (3.8%)	96 (3.2%)	365 (3.3%)
Physical features	Dominant hand						
		Left	84 (3.9%)	93 (3.2%)	112 (3.8%)	102 (3.4%)	391 (3.5%)
		Right	2008 (92.2%)	2675 (92.7%)	2729 (91.4%)	2758 (92.3%)	10,170 (92.2%)
		Both	86 (3.9%)	117 (4.1%)	145 (4.9%)	127 (4.3%)	475 (4.3%)
	Height (cm)						
		Mean (SD)	164 (8.42)	164 (8.30)	163 (8.15)	163 (8.24)	163 (8.28)
		Median [Min, Max]	163 [141, 194]	163 [140, 194]	162 [141, 192]	162 [139, 193]	162 [139, 194]
	Weight (kg)						
		Mean (SD)	63.5 (12.5)	63.9 (12.8)	63.3 (12.1)	63.2 (12.0)	63.5 (12.3)
		Median [Min, Max]	61.0 [37.7, 119.9]	61.5 [37.7, 152.0]	61.1 [38.3, 130.6]	61.1 [32.0, 137.8]	162 [32.0, 152.0]
Habits	Smoking						
		Yes	442 (20.5%)	626 (21.9%)	551 (18.5%)	511 (17.1%)	2130 (19.4%)
		No	1714 (79.5%)	2231 (78.1%)	2435 (81.5%)	2474 (82.9%)	8854 (80.6%)
	Drinking						
		Yes	1489 (69.1%)	1883 (65.9%)	1918 (64.2%)	1814 (60.8%)	7104 (64.7%)
		No	665 (30.9%)	974 (34.1%)	1068 (35.8%)	1171 (39.2%)	3878 (35.3%)

## Data Availability

The data presented in this study are available upon request from the corresponding author. These data are not publicly available because of privacy concerns.

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
