# Peer review of "Establishment of a Dataset for the Traditional Korean Medicine Examination in Healthy Adults"

_healthcare, 2024, doi:10.3390/healthcare12090918_

Round 1

Reviewer 1 Report

Comments and Suggestions for Authors

In the present manuscript, the authors suggested the SOP, questionnaire to collect the information about the patient who want to go for KM. They have compiled the data from different hospitals/ health centres. Here are some of my observations which needs justification:

- Line no 306, authors stated that they methodically gather disease specific patient data, however in the manuscript, disease specific sop, and examinations/ test are not specified. 
- SOP suggested seems to be common for the evaluation of any disease in any  alternative system of medicine. The traditional practioner, generally follows the particular sop, questions or tests on the basis of symptoms of patients. So what is the importance of compiled information, if it is not disease specific and if it is so then also it is mostly decided by practitioner to go for which test or sop. Why to follow for all test. 
- Impact of questions asked and sop is not compared with disease person or not quantified so how importance of compiled information can be surely justified.

- The authors concluded that information of about 11000 person is digitised can be used for future programs. Were the patients informed about the future use of their personal information for future programs. 

Comments on the Quality of English Language

Minor

Reviewer 2 Report

Comments and Suggestions for Authors

Interesting manuscript, The results can contribute to improve processes and to develop accurate database with health information.
Please add some information about the state of the art of the area . Any previous study that shown similar results. How has the consistency of the study  with the chosen methodology and with previous publications been analyzed ?

Not totally clear if this study is a case study or a qualitative study in which a pilot study has been performed . Please clarify.  

Reviewer 3 Report

Comments and Suggestions for Authors

See enclosed comments.

Comments on the Quality of English Language

See enclosed comments.

Round 2

Reviewer 1 Report

Comments and Suggestions for Authors

Nil

Comments on the Quality of English Language

Minor editing required

Author Response

Thank you for the positive answers. This manuscript has undergone a comprehensive review and correction process by a professional editor. He has meticulously examined the content for clarity, coherence, and accuracy. Furthermore, he has ensured that the manuscript adheres to the appropriate style and formatting guidelines.

Reviewer 3 Report

Comments and Suggestions for Authors

See enclosed comments.

Comments on the Quality of English Language

See enclosed comments.
